# A Compendium of G-Flipon Biological Functions That Have Experimental Validation

**DOI:** 10.3390/ijms251910299

**Published:** 2024-09-25

**Authors:** Alan Herbert

**Affiliations:** Discovery, InsideOutBio, 42 8th Street, Unit 3412, Charlestown, MA 02129, USA; alan.herbert@insideoutbio.com

**Keywords:** G-quadruplex, flipons, transcription, splicing, translation, recombination, methylation, R-loop, chromatin loop, telomere CTCF, helicase, HIV, bistable switch, Rap1, Z-DNA

## Abstract

As with all new fields of discovery, work on the biological role of G-quadruplexes (GQs) has produced a number of results that at first glance are quite baffling, sometimes because they do not fit well together, but mostly because they are different from commonly held expectations. Like other classes of flipons, those that form G-quadruplexes have a repeat sequence motif that enables the fold. The canonical DNA motif (G_3_N_1–7_)_3_G_3_, where N is any nucleotide and G is guanine, is a feature that is under active selection in avian and mammalian genomes. The involvement of G-flipons in genome maintenance traces back to the invertebrate *Caenorhabditis elegans* and to ancient DNA repair pathways. The role of GQs in transcription is supported by the observation that yeast Rap1 protein binds both B-DNA, in a sequence-specific manner, and GQs, in a structure-specific manner, through the same helix. Other sequence-specific transcription factors (TFs) also engage both conformations to actuate cellular transactions. Noncoding RNAs can also modulate GQ formation in a sequence-specific manner and engage the same cellular machinery as localized by TFs, linking the ancient RNA world with the modern protein world. The coevolution of noncoding RNAs and sequence-specific proteins is supported by studies of early embryonic development, where the transient formation of G-quadruplexes coordinates the epigenetic specification of cell fate.

## 1. Introduction

The idea that the repetitive genome encodes genetic information by shape rather than by sequence is relatively new. The unit of information is the flipon, a genomic element that can adopt alternative structures under physiological conditions. The conformation formed depends on the repeat sequence involved. The classic example is provided by left-handed Z-DNAs and Z-RNAs (collectively called ZNAs) that are formed by runs of alternating guanosine and cytosine [1,2]. Collectively, the repetitive genome comprises over 50% of the human sequence, compared to 2.5% for protein-coding genes.

Flipons in the B-DNA conformation have little informational value, as the repeats are frequent in the genome. They also lack the complexity of codons, so they do not contribute directly to the Watson and Crick genetics that focuses on protein variation. Instead, flipons alter the readout of genetic information by localizing structure-specific complexes to genomic loci able to power the flip from a right-handed B-DNA or A-RNA helix to an alternative DNA or RNA fold. The readout of RNAs then varies dynamically with flipon structure. Here, the focus is on G-flipons that form G-quadruplexes (GQs) in DNA (dGQs), RNA (rGQs) or DNA/RNA hybrids (hGQs). GQs are inherently more stable than ZNA helices. Consequently, G-flipons can actuate biological processes that are quite distinct from those modulated by Z-flipons.

GQ-forming sequences are defined by the canonical DNA motif (G_3_N_1–7_)_3_G_3_, where G is guanine and N is any nucleotide. Four G-bases hydrogen-bond to each other to form a tetrad that then folds into a four-stranded structure. In place of the Watson–Crick base-pairing scheme, the rather unconventional Hoogsteen hydrogen bonds stabilize the interaction. The G-tetrad was first observed in X-ray diffraction studies of 5′-GMP and 3′-GMP gels, each stacking the tetrads on top of one another differently [3]. The preferred helical arrangement of GQ crystalline fibers was later revealed by structural studies of polyinosinic and polyguanylic RNAs [4].

It was once widely believed that GQs did not exist in cells. If present, then the GQs formed were predisposed to genetic instability and to disease [5]. There was much excitement when the Tetrahymena telomere sequence repeats [6] were shown to form GQs [7]. In contrast, later work revealed that telomeres in vivo were more likely to form a different type of structure called a T-loop [8]. The closure of the loop led to the formation of a three-stranded DNA structure that incorporated the single-stranded telomeric end and a subtelomeric segment. This structure was protected by a shelterin protein complex. The T-loop model seemingly ruled out a role for GQs in telomere maintenance (but see below). The prevailing view that GQs were bad was reinforced by the many loss-of-function (LOF) helicase variants that were associated with human mendelian diseases. The failure of these variants to resolve GQs was considered causal for genomic instability, even though the helicases also resolved other non-B structures, such as cruciforms and Holliday junctions (HJs) that form during recombination [9]. Further, a role for GQs in pathology was suggested by an analysis of repeat expansion diseases. In some cases, the sequences involved were predicted to freeze in the GQ conformation, thereby interfering with a variety of cellular functions, including DNA replication, transcription and RNA processing [10].

However, there was evidence that GQs played an essential biological role in the adaptive immune system. The GQs were associated with the class switch recombination of immunoglobulin heavy chain (*IgH*) genes. Of interest were the noncoding switch (S) regions in the *IgH* gene that underwent transcription to produce R-loops. The non-template strand was G-rich and 2 to 10 kb in length. When displaced by RNA transcripts, the single-stranded G-rich DNA was able to fold back on itself to form GQs [11]. The targeting of the AID cytosine deaminase protein to the GQ structure by helicase DDX1 was essential for both class switching and the immunoglobulin somatic hypermutation that is critical for antibody affinity maturation [11,12,13,14]. The cytosine-to-uridine substitution catalyzed by the cytidine deaminase was not only mutagenic but also recruited the repair machinery required for DNA recombination. In other contexts, GQ formation in G-rich DNA due to R-loop formation was proposed as pathogenic [15].

Other experimental approaches to unraveling the biology of GQs were complicated by the equilibrium that exists between different flipon conformations, with the transition occurring in unmodified DNA and without requiring any strand cleavage [1]. Early experiments using dimethyl sulfoxide footprinting of RNA failed to show the protection of guanine bases expected if a GQ had formed inside a cell [16]. These results were interpreted to show that GQs were not biologically relevant. However, there was a problem with the experimental design: chemical modification of any G-quadruplexes that unfolded during the time course of the experiment would prevent the structure from reforming [17]. In other words, the longer the experiment ran, the less chance there was of detecting the presence of GQs in a cell. Nevertheless, the study highlighted the possibility that GQs were formed dynamically in cells and that they were rapidly resolved to reform B-DNA.

There were also limitations to other experimental approaches designed to detect GQs. Tools designed to detect GQs in cells were able to induce their formation. This risk of an artifact increased when assays were performed on cell extracts. Here, various factors came into play, such as the buffers used and the loss of proteins that might otherwise restrain the B-DNA flip to GQs. Even well-accepted ChIP-seq protocols to map protein interactions are potentially misleading, as recently shown by a stringent analysis of the interactions between the GQ-binding substrates of PRC2 (Polycomb Repressor Complex 2) [18]. Combined, these uncertainties limited the widespread acceptance of G-flipons as important components of the genetic repertoire. The repetitive genome was just considered “junk” [19].

The intent of this review is to integrate information from a wide range of research papers, including some whose significance has long been overlooked and who are not mentioned in many recent GQ reviews [20,21,22,23,24,25,26,27,28]. The initial focus is on the genetic evidence that speaks to an early evolutionary role for G-flipons in maintaining genomic stability and on the proteins that localize the machinery required for nucleotide and base excision repair (NER and BER, respectively) by inducing GQ formation. Different classes of helicase then power the resolution of GQs to reform B-DNA, completing the flipon cycle. By changing the readout of genetic information, flipons dynamically reprogram a cell in response to environmental perturbations.

I will then discuss the roles of G-flipons in transcription that emerged later in evolution. This feature reflects a change in how GQ recognition occurs, from interactions involving single-stranded DNA loops and modified bases to those mediated by proteins that bind both B-DNA and G-quadruplexes through a different face of the same helix.

## 2. Biophysical and Computational Studies of the G-Quadruplex

The basic building block for a G-quadruplex is a guanine tetrad formed by Hoogsteen hydrogen bonding [4] (Colored in Figure 1B) between bases [29]. Interestingly, the parallel nature of these bonds contributes to sigma bonds, which increase the stability of the G-tetrads relative to those formed by xanthine, where the bonding is anti-parallel [30]. A recent review describes 48 different possible GQ folds, reflecting whether the four strands are parallel, anti-parallel or a mix, made from one to four different strands with a lateral, diagonal or propeller loop topology [31] (Figure 1). Further, the guanosine residues may be either in a *syn* or *anti* conformation (with the guanine base either lying over the sugar or pointed away from it) [32]. The GQs could also be left-handed [33]. The folds are stabilized by a central metal, with a potassium ion preferred over the smaller sodium and lithium ions for parallel-strand GQs. The metal preference for other GQ folds varies and depends on whether they are made from RNA or DNA [34]. Non-consecutive guanosines can form tetrads with the extra residue everted from the stack to form a bulge. In the case of GGA repeats, the adenine bases that are excluded from the quadruplex can interact with the tetrads to produce a heptad structure [35,36]. 

The stability of GQs is also affected by the loop composition, decreasing with loop length, and varying with the loop nucleotide sequence [38]. With long runs of G-repeats, defined as over 500 bases in length, the loops can base-pair to give even higher-order structures. Of the 299 long G runs reported, over 67% are located within 6M bp (base pairs) of telomeres [39]. Interestingly, GQ loop length and sequence variation have increased during evolution, especially in mammals, and as has GQ length, number, and density in the genome [40]. G-flipons are also more frequent on the non-template strand of coding genes [41,42].

Besides GQ formation by neighboring G3 repeats, it has been proposed that GQs are formed by a pair of G3 repeats in an enhancer and a pair of G3 repeats from a promoter [43]. Further, a hybrid GQ can form between a pair of DNA G3 repeats in the non-template strand and a matching RNA G3 pair in the nascent transcript [44]. The GQs formed by strands that are not physically connected to each other also show structural variation. The GQs can assemble by stacking tetrads one on top of the other or by pairing bases from the separate strands to form a G-wire [45,46]. G-wires were originally proposed to explain the alignment of homologous chromosomes during meiosis [47]. Tetrads missing their fourth base can incorporate into the vacant space a guanine provided *in trans*, potentially acting as a sensor for a local change in the concentration of the replacement nucleotide [48].

RNA tetrads only form parallel rGQs when G-repeats are contiguous [34]. A variety of non-canonical rGQ folds are stabilized by pairing of G-bases that are widely separated by non-G nucleotides [49]. rGQs composed of only two tetrads have been reported [50] and are stabilized by the 2′-hydroxyl groups present in RNA [34]. In contrast, there are many possible variations in dGQs composed of three or more tetrads, making it difficult to computationally predict from sequence alone the flipons that actually form dGQs in vivo. A database that combines results from a variety of experimental methods now overcomes this problem by providing a set of well-validated G-flipons detected in many different studies using a variety of approaches [51]. The mappings show that in the human genome, dGQ-forming sequences are enriched in transcription start sites (TSSs), in introns and at transcription termination sites (TTSs) [40].

## 3. GQ-Binding Proteins

The plethora of different dGQ topologies allows for different modes of protein recognition (Figure 2 and Figure 3). Strategies to confirm these interactions and the specificity of binding to GQs include those that synthesize control oligonucleotides containing an 8-aza-7-deazaguanosine base (Figure 1C) that will not form the Hoogsteen hydrogen bonds necessary to stabilize a GQ (Figure 1B, crimson shading) despite having the same chemical composition as guanine [52]. In these studies, different modes of docking to GQs have been identified, including binding to loop sequences or to 5′ and 3′ single-strand extensions that give the helicases something to pull on so that they can unwind their structure. Proteins can bind to loops formed when adducted bases such as 8-oxo-G prevent the incorporation of a DNA strand into a GQ, or to the everted bases across from an apurinic/apyrimidinic (AP) site. Proteins also dock to the planar tetrad surfaces that form the GQ endplate. Specific binding to rGQs rather than dGQs is favored by intrinsically disordered regions (IDRs) enriched in arginine and by glycine repeats, as recently reviewed [53], and as visualized in the FMR crystal structure of the Fragile X Mental Retardation Protein bound to an rGQ [54]. In principle, the preformed GQ site for docking IDRs lowers the entropic cost of binding.

The stability of GQs and strength of their interactions with proteins can vary with the loop length and loop sequence composition [57,58], as revealed by studies of nucleolin and the 2E4 Darpin [59,60]. Further, the latching of a single base by the REV1 polymerase [61], and the docking to an AP site by APE1 (AP endonuclease 1) [62], can create a surface that induces GQ folding. As we will discuss, the use of SANT (Swi3, Ada2, N-Cor and TFIIIB) domains to recognize parallel-strand GQs is of particular interest, as the domains can use the same helix to bind B-DNA in a sequence-specific manner. In total, 50 GQ–peptide structures are present in the Protein Database (PDB), showing a variety of interactions [26,60]. A subset of validated GQ-interacting proteins is given in figures below. Listings of additional proposed GQ-binding proteins can be found in recent publications [52,63], online in the G4IPBD database (http://people.iiti.ac.in/~amitk/bsbe/ipdb/index.php, accessed on 15 September 2024) [64] and the QUADRatlas database (https://rg4db.cibio.unitn.it/, accessed on 15 September 2024) [65].

## 4. The Accumulating Evidence for the Biological Importance of G-Quadruplexes

Despite the numerous challenges to studying the cellular functions of high-energy and dynamic flipon conformations, much progress has been made. There are two key aspects to their biology: first, the events that promote and resolve the formation of the alternative flipon structures, and second, the transactions that the alternative flipon conformations actuate. There are well-validated proteins that can induce the flip to GQs and many helicases capable of their resolution (Figure 2, Figure 3, up and down arrows). Although GQ formation does not inherently require any change, modification or cleavage of DNA or RNA, such events may change the propensity of G-flipons to flip from one conformation to another. The GQs formed in these processes differ in topology. The structured loops they form are recognized by specific sets of proteins; as are the GQ endplates (Figure 3, top). The outcomes depend on which cellular machinery is localized to a particular GQ. The complexes formed enable cells to reprogram their responses to environmental perturbations.

The transactions occurring between GQs formed at different sites are also important in understanding their cellular functions. The complexes nucleated by one GQ have the potential to associate with other G4-anchored structures to form membraneless condensates (Figure 4) [66,67]. These complexes can be quite large and visible by light microscopy [66]. Their interactions enable the sequencing and timing of events within the cell (Figure 4A). The pairings of promoter GQs with GQs formed at enhancers, splice sites and polyadenylation sites then generate production lines for the processing of transcripts. Factories form by anchoring of the production lines to the nuclear scaffold [68,69], delivering the transcriptional bursts associated with gene expression [70]. The pliability of these production lines is revealed by constant updates to the nuclear architecture [71]. G-flipons actuate many outcomes in the cell and are exploited by pathogens, as described below.

### 4.1. Retroviral Latency

The simplest example of GQ-mediated integration may be provided by retroviruses, such as human immunodeficiency virus 1 (HIV-1). These viruses encode G-flipons in the long terminal repeat that are present at either end of their 9.6 kb genomic insert [72] (Figure 4B). This arrangement enables the formation of chromatin loops that separate the viral protein-coding genome from that of the host. In this state, the virus is likely latent. Nevertheless, the virus is poised to replicate upon the removal of the loop restraint (Figure 4B). The HIV-1 plus-strand mRNA also contains 11 potential G-quadruplexes, with 9 in the coding sequence. The topologies are mixed, raising the possibility that particular pairings affect the splicing, stability, recombination, and repair of viral transcripts [73]. Long interspersed elements (LINEs) are another class of retrotransposons that have a G-flipon conserved in their 3′UTR (untranslated region). The pairing of LINE GQs with GQs in cellular enhancers also has the potential to form a loop that controls their expression in a tissue-specific manner [74]. Conversely, the 5′UTR G-flipons that LINE families acquire during evolution can themselves act as tissue-specific enhancers for cellular genes [75].

### 4.2. Cell Division

Interestingly, the first evidence hinting at a biological role for GQs came from the round worm *Caenorhabditis elegans*. Sequences with G-quadruplex motifs underwent deletion in strains with *dog-1* (deletions of guanine-rich DNA) LOF variants, but not in sequences with only three G_3_ repeats that are unable to form GQs [76]. Mutant strains of *dog-1* lacking the trans-lesion (TLS) polymerases POL eta and POL kappa had significantly more G-tract deletions than dog-1 by itself [77]. Interestingly, the combined deletion of *dog-1* and the spindle-checkpoint component *mdf-1* enabled long-term survival [78], even though a high incidence of lethal mutations in this strain was revealed by the use of balancer chromosomes. In total, 126 (13%) of the 954 mono-G/C tracts larger than 14 bp were deleted over 470 generations when both genes were absent. A role of GQs in sister chromatid alignment by the cohesin proteins during mitosis was suggested by the effects of *dog-1* LOF on the spindle checkpoint. The absence of other phenotypes also supported the consensus that GQs had only a limited role in normal cell biology, not only in *C. elegans* but also in other organisms.

### 4.3. Epigenetic Maintenance

A *dog-1* homolog in the DT40 chicken lymphoblastoid cell line, 5′ FANCJ (Fanconi Anemia Complementation Group J) helicase (a member of the Fe-S superfamily 2 (SF2)) [79] was also found to prevent the deletion of guanine repeats (G-repeats) that have the potential to form GQs. The effects of the mutation were enhanced by the loss of REV1 polymerase, which localizes TLS polymerases to sites of polymerase stalling. Interestingly, REV1 catalytic activity was not necessary to prevent deletion, although the LOF variant did enhance the rate of G-repeat loss. Also, in the FANCJ model, the combined deletion of the Werner and Bloom Syndrome 3′ helicases (RecQ SF2) [80] also increased G-repeat deletion, likely because of GQ accumulation [79].

Of interest is that the TLS pathway was required to maintain the epigenetic state of dividing cells, as monitored by the cell-surface expression of a protein with an intronic G-flipon that regulated gene expression. In contrast, in the wildtype cell, the histone modifications associated with this G-flipon were maintained; they were lost following *rev1* deletion. Instead, the resolution of the GQs formed during DNA replication occurred through the gap-filling repair pathway. The subsequent incorporation of unmodified histones led to diminished gene transcription and surface marker expression. This *rev1*-dependent phenotype could be reverted by re-expression of human FANCJ helicase [79]. The opposite effect was observed when a G-flipon was experimentally inserted into a repressed locus. In this case, *rev1* deletion led to the depression of the segment, consistent with the replacement of repressive histones with unmodified histones that were permissive to gene expression [81]. These results support a model where the formation of GQs by G-flipons during periods of cell proliferation helps in transmitting the current epigenetic state to progeny, an important biological outcome.

### 4.4. DNA Replication and Sister Chromatid Conformation

The involvement of GQs in cell proliferation is further supported by other evidence. During assembly of the DNA polymerase complex at the origin of replication (OOR), the MTBP protein assists in the loading of CDC45 into the replicative helicase. The C-terminal domain of MTBP binds GQs in vitro [82]. Notably, G-flipons are enriched in OOR. Indeed, in chicken DT20 cells, a minimal, functional OOR consists of a 90 bp fragment that has two G-flipons on the same strand [83]. These constructs establish the nucleosome-depleted region (NDR), which is bound by histone H2A.Z and is typical of the OOR. Collectively, the results suggest a model in which the MTBP binds GQs at the OOR to initiate the assembly of the replication complex.

Another potential role for GQs during the proliferation and transmission of the epigenetic state is to align sister chromatids, as the mapping of intra- and inter-chromatin interactions between homologous chromosomes reveals a high degree of symmetry in the architecture of topologically associated domains (TADs), and in the loops formed within TADs [84]. In this regard, a recent report suggests that G-flipons are enriched near sites bound by the CTCF (CCCTC-binding factor), a protein associated with loop formation. Interestingly, the strand orientation of the G-flipons mirrors the inverse orientation of the two CTCF sites that associate with each other to form the base of the chromatin loop [85]. CTCF, however, is not known to bind GQs [52]. 

### 4.5. Nucleotide Excision Repair (NER)

The REV1 pathway also plays a role in NER, which is triggered by UV irradiation and the formation of DNA crosslinks. In this situation, the loading of repair pathway proteins such as XPCC and RAD23 is triggered by the protein ZRF1 and its yeast homolog Zuotein, which recognizes the lesion and induces GQ formation [86]. Triggering this pathway by cytosine deaminases can result in single-base substitutions at a sequence-tagged site (STS), with a C to G transversion resulting from the preferential insertion of cytidine into the lesion by REV1 [87]. The resulting mutation (STS13) is prevalent in cancers [88].

NER in the transcription-coupled repair pathway (TCR) depends on the Cockayne Syndrome B (CSB) helicase (encoded by ERCC6) that binds GQs [89]. On sensing a lesion, CSB displaces DSIF (DRB Sensitivity Inducing Factor) from the RNA polymerase 2 (RNAP2) complex, inducing a conformational switch that halts transcriptional elongation and initiates TCR [90]. LOF variants of CSB are associated with premature aging phenotypes [89].

### 4.6. Base Excision Repair (BER)

APE1 plays a similar role in stabilizing GQs formed by AP DNA, but not unmodified DNA, to initiate the BER pathway [62]. This pathway removes oxidized bases, such as 8-oxo-G. It is proposed that the regulation of APE1 by acetylation coordinates the expression of genes involved in cellular pathways that respond to oxidative damage. Interestingly, the GQs involved are formed from G-flipons with a “spare tire” (Figure 1F). The extra runs of G-repeats allow the formation of a GQ despite damage to one of the other repeats [91]. 

The 8-oxoG modification can arise due to toxins in the environment. The adduct is also generated routinely during the flavin-dependent LSD1 (lysine demethylase 1A, encoded by KDM1A) demethylation of H3K9me2, where hydrogen peroxide is a product of the reaction. The LSD1 enzyme is activated during the induction of BCL2 gene expression by estrogen [62]. The repair of the lesion through the BER pathway depends on GQ formation. Before the involvement of GQs in this process was known, it was proposed that the DNA strand breaks observed were a general mechanism for initiating gene transcription [92], but now can be viewed as just another example of how flipons enable the reset of chromatin.

### 4.7. Hemin and Oxidative Damage

Oxidative damage also arises from the production of highly reactive oxidative species catalyzed by hemin, an iron-containing porphyrin that is present at high concentrations in the cell [93]. Hemin binds with high affinity (K_d_ ~ 10 nM) to GQs, an interaction that was initially highlighted for its ability to increase the production of superoxides [94]. However, it appears that in cells, this reaction is squelched, presumably by proteins that bind to GQs [93]. In such cases, GQs may act as a sink for free hemin and trigger the rapid repair, through the BER pathway, of the damaged bases produced. In such cases, GQs protect, rather than damage, the genome.

### 4.8. Telomere Protection

The formation of T-loops by telomeres described above does not rule out the role of GQ formation in telomere protection. Indeed, the GQ binding TRF2/RAP1 (telomere repeat binding factor 2/ repressor-activator protein 1) complex protects telomeres from homologous recombination by repressing PARP1 (poly(ADP-ribose) polymerase 1) localization to telomeres and by inhibiting the SLX4 resolvase that binds to HJs. The loss of TRF2 and RAP1 in both humans and mice leads to rapid telomere attrition, with increased rates of telomere deletion and fusion [95]. TRF2 preferentially docks to rGQs rather than dGQs. The protein binds rGQs formed by the noncoding Telomeric Repeat-Containing RNA (TERRA) telomere transcript through an RG-rich domain [96]. Interestingly, the HIV-1 retrovirus may form a dGQ to cap the DNA flap sequence produced during the pre-integration phase of reverse transcription, potentially protecting the end in much the same way as proposed for host telomeres [97].

### 4.9. Resolution of G-Quadruplexes

Implicit in the G-flipon cycle is the need to reset flipons to a resting state. As shown in Figure 3, many helicases enable this outcome. The most studied example is the ATP-dependent DEAH box SF2 helicase DDX36 (RHAU), a highly specific GQ resolvase that unwinds parallel dGQs. The enzyme makes helical contact with the GQ end plate [98,99]. Binding by the helix alone has a relatively high K_d_ of 1 μM. The additional engagement of a 3′ single-stranded dGQ tail by other residues accounts for the nM affinity of the enzyme for its substrate. Using a ratchet mechanism, the helicase disassembles the dGQs, one guanine at a time. The chemical energy derived from ATP is converted into a pulling force by rotation of the C-terminal domain. The twist opens up the helicase core [99]. In the absence of nucleotides, or in the ADP-bound state, *D. melanogaster* DDX36 stabilizes GQs [100].

The cocrystal structure of dGQs with the SF1 *Thermus oshimai* 5′-3′ Pif1 helicase shows the enzyme in an unwinding state with the engagement of a single-stranded thymine repeat [101]. The related yeast helicases *PiF1* and *Rrm3* cooperate to unfold a wide range of dGQ topologies, including those formed not only by telomeres, but also by centromeres and tRNA repeat sequences [102,103]. The enzyme unfolds dGQs in an ATP-dependent manner, unwinding both parallel and antiparallel dGQs [101]. The interaction of the Pif1 with a parallel-stranded dGQ differs from that with DDX36. The contact is mediated by a cluster of amino acids, including two arginine/lysine cation–π interactions at either end of the dGQ, plus ionic contacts with the phosphate backbone. The SF2 RecQ BLM helicase also unfolds a variety of dGQ through a number of different mechanisms [104]. Collectively, the helicases play key but distinct roles in flipping dGQs back to B-DNA.

### 4.10. G-Flipons and Gene Expression

The widely held assumption is that a crystal structure of a protein engaged with B-DNA precludes an interaction with any other DNA conformation, especially if the substrate is bound with nM affinity. Of course, crystal structures by their nature represent a low-energy state. The example of Rap1 is therefore instructive (Figure 2). Prior to its role in telomere protection, Rap1 was characterized as a sequence-specific transcription factor that was bound to a UAS (upstream activating site) in yeast [105]. Its base-specific interaction with B-DNA was confirmed by a crystallographic study of a telomeric sequence (Figure 2A) [55]. Only later did crystal structures show that Rap1 also docked to GQs. Surprisingly, both DNA interactions involved the same helix, but with a different face [56] (Figure 2B). The GQ contacts were hydrophobic, with the helix lying on the planar surface of the terminal tetrad, while the B-DNA contacts were consistent with those found for the UAS. Both interactions have a K_d_ ≈ 20–30 nM [56], yielding a switch that has two stable states (Figure 2C). The switch state then depends on the context and the availability of the helicases. The example illustrates the potential of flipons to switch the readout of genetic information from a genome by changes to their conformation [106].

While this finding might seem anomalous, many subsequent studies have demonstrated the ability of proteins to both bind specifically to a cognate B-DNA sequence, and also to a GQ, often with nanomolar affinity for both conformations. This finding is true for the binding of the SP1 transcription factor to its B-DNA cognate sequence and a c-MYC parallel GQ [107] and many other proteins that bind both to GQs and to a B-DNA motif [21]. Interestingly, like Rap1, many of the GQ-binding proteins include a SANT/Myb domain such as ZRF1 [108] and TRF2 [109,110]. Interestingly, the yeast Zuotin protein has replaced the SANT domain with a highly hydrophobic helix that could well interact with a GQ endplate [108]. SANT-domain proteins are found in multiple chromatin-modifying and remodeling complexes, although their interactions with GQs are not yet reported [111].

### 4.11. Enhancer Promoter Condensates 

Given the enrichment of G-flipons in promoters, a key question was how do proteins that stabilize and resolve GQs impact transcription. GQ-binding proteins like YY1 (Yin Yang 1) are known to form homodimers that promote enhancer–promoter contacts [52,112,113]. So do transcription factors that bind GQs. One of the surprises of the ENCODE project was the identification of HOT (high occupancy target) loci where upwards of a 100 TF bound, even to sites lacking their sequence-specific binding motif. The findings were initially dismissed as methodological artifacts [114], but were later shown not to be so [115,116]. The primary studies focused on the sequence-specificity of TFs, not on the GQs that were also formed at promoters. The ability of TFs to bind both B-DNA and GQs offered a resolution to this HOT dilemma [52]. Indeed, recent findings suggest that it is GQ formation that recruits TFs to transcriptional hubs [117]. In this new model, as described here, TFs play a non-traditional role. Through the complexes they anchor, TFs localize helicases to resolve the GQs formed by promoters. A specific helicase might recognize a particular GQ fold or a GQ loop of a particular length or composition, or display a preference for a 5′ or 3′ single-stranded flanking sequence. The biological outcomes then depend on the GQ’s topology and the helicase involved. G-flipon cycle are then able to actuate a diverse set of transactions (Figure 3).

### 4.12. Transcriptional Bursting

One extension of this model is that the docking of TFs to GQs maintains a constant state following the initiation of transcription by the binding of a sequence-specific TF to B-DNA. Consequently, there would be no need for any further sequence-specific interactions with the promoter. However, this possibility is not consistent with the observed rapid reset of promoters that occurs after each round of transcription [118,119]. The fast disassembly of the transcriptional complexes following each round of transcription is mirrored by the abrupt dissolution of promoter condensates, triggered by the high levels of nascent RNAs produced [120]. This evidence suggests that transcription occurs in bursts followed by a reset, rather than by a constant, preset rate of expression.
Figure 4**G-flipon nucleate condensates**. In the scheme presented, GQ formation seeds condensates that promote transactions between different genomic regions. This arrangement can maintain G-flipons in an active but poised state, locked and loaded, ready to actuate a particular outcome. (**A**) The sequential contacts between the GQs formed at different chromosomal sites ensure that RNA processing occurs in the correct temporal order. The splice and polyadenylation sites selected may vary with the specific promoter used to seed the condensate. The GQ folds formed may vary by site. The dotted lines indicate that both DNA and RNA GQs can participate in the transactions. (**B**) Retroviruses are enriched for G-flipons in their LTR, which can adopt different conformations, as seen in the NMR structures PDB:2N47 [121] and PDB:6HiK [122]. The formation of GQs by both retroviral LTRs may anchor chromosomal loops that are stabilized by a condensate at their base. In this state, the viral genome is poised, but not actively transcribed. The dissolution of the condensate then releases RPOL2 to initiate transcription. The viral plus strand has eleven conserved sequences capable of forming different GQ, folds, nine of which are in protein-coding regions. Two potential GQs are present on the negative strand [73,123,124]. The number of potential G-flipons in the LTR differs between retroviruses [72]. Whether the presence of more GQs increases viral virulence or whether the G-flipons enable different processing events is not currently known.
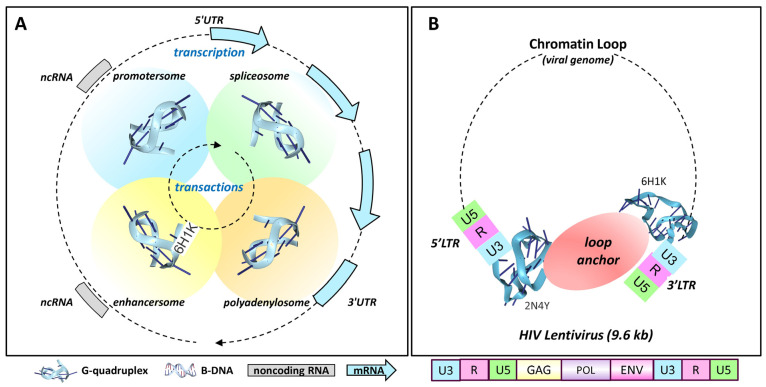



Earlier experiments based on single-molecule FISH suggested that the transcriptional burst frequency, but not the burst size, depended on the rate of promoter reset [118]. One contribution to burst size was the frequency with which sister chromatids were transcribed. Curiously, only one allele was active at a time, rather than both undergoing simultaneous transcription [118]. The localization of many different helicases to the locus might then allow one allele to reload a sequence-specific TF to reform an initiation complex while the other one fired. Such coordinated activity is supported by the symmetrical chromatin architecture observed for sister chromatids, as described above [84]. The lack of co-bursting by maternal and paternal chromosomes is consistent with recent single-cell studies of allele-specific transcription [125].

### 4.13. Promoter Pausing

How, then, do GQs modulate transcriptional bursting? The formation of GQs at promoters is often detectable before the initiation of transcription [126]. In such situations, there is no preference for which strand forms a GQ, further suggesting that gene transcription is not required to flip B-DNA to the GQ conformation [107]. Further, the GQ flip is not modulated by human topoisomerase I (TOP1), even though the enzyme is enriched at these sites [127]. Instead, TOP1 is inhibited by GQs, with an IC50 ~ 100 nM [128]. 

It is possible that the GQs formed at promoters engage RPOL2, but prevent elongation by holding the enzyme in a poised state (Figure 5A). The YY1-mediated looping between promoter and enhancer GQs could further freeze RPOL2 in place through the condensate formed [52,112,113]. Locking down RPOL2 then provides the time to properly position other GQ-anchored condensates required to correctly splice and polyadenylate the pre-mRNA produced. Without the proper arrangements in place to coordinate downstream events, an elongating RPOL2 will terminate transcription prematurely and detach from the DNA template (Figure 4) [129]. Following the release of RPOL2 from the TSS, the enhancer–promoter condensate undergoes disassembly, enabling the promoter to reset for another transcriptional burst [120].

### 4.14. G- and Z-Flipons and Promoter Reset

This scenario provides a different perspective on why TF engagement in promoter GQs is important in regulating gene expression. In the new scheme, TFs do not directly drive gene expression by binding a cognate motif. Instead, they engage GQs, localizing complexes that contain the helicases required to rapidly resolve GQs. The reset then allows the DNA duplex to reform and B-DNA sequence-specific proteins to seed the formation of a new promoter/enhancer condensate.

It is also necessary to clear the existing pre-initiation complex (PIC) used previously to dock RPOL2 at the transcription bubble (Figure 5). This process often involves Z-DNA-forming sequences near the TSS. Here, the negative supercoiling generated by an elongating RNAP2 powers the formation of Z-DNA [130,131,132]. The energy stored in Z-DNA then actuates the removal of the existing PIC from the promoter. A new PIC more in tune with the current state of the cell can then form (Figure 5C) [130]. The formation of Z-DNA may also be necessary to reengage the RPOL2 complex. There is preliminary evidence that GTFE (General Transcriptional Factor E) binds to Z-DNA, followed by docking of RPOLR2 to a newly formed transcription bubble [130,131]. 

The reengagement of the RPOL2 complex also depends on the binding to the promoter of sequence-specific TFs to B-DNA that help seed the PIC. The positive supercoiling induced by PIC engagement [132] then modulates the conformation of both G- and Z-flipon. The positive supercoiling will lead to the unwinding of DNA on either side of the PIC [133]. The upstream uncoiling of DNA will promote GQ formation, signaling that the PIC is engaged, while the downstream unwinding will assist in opening the RNAP2 catalytic center allowing the coding strand to enter (Figure 5C).

The rest and reinitiation mechanism based on flipons is likely quite ancient. Indeed, many of the promoters regulating embryonic and neurological development contain both G- and Z-flipons that have been validated experimentally [134]. This mechanism is quite flexible and adaptable. The insertion of flipons during evolution into promoters by retrotransposons provides a mechanism for modulating gene expression. In humans, the copying and pasting of the ALU family of SINEs (short interspersed nuclear elements) throughout the genome has greatly enhanced this type of genetic variation [106]. An extreme example of the alternative outcomes enabled by the insertion of flipons into promoter regions is provided by the experimental observation that some flipons can form either GQs or Z-DNA [134,135]. The particular structure adopted may depend on which TSS is used to initiate gene expression. The formation of Z-DNA downstream would be driven by transcription, while the flip upstream to GQs would be driven by TF engagement.

The involvement of G-flipons in both polymerase pausing and in promoter reset may produce some paradoxical results when ligands that stabilize GQs are employed experimentally. The outcome then depends on the step in the G-flipon cycle that is most affected. The immediate effect of disrupting the GQ-dependent enhancer–promoter condensate is the release of RPOL2 from the promoter and a transcriptional burst. Failure to reset the promoter will maintain the NDR and increase the chance of DNA damage, leading to decreased transcription initiation and eventually to cell death. It is also possible that a GQ-stabilizing ligand may disrupt the reset of GQs at genomic sites other than the promoter, leading to premature transcript termination due to the disruption of downstream RNA processing events.

### 4.15. Gene Repression

*GQs and gene repression*. The promoter reset occurs in competition with protein that suppress gene expression. These competitors include the PRC2 complex that engages the GQs formed at promoters through the SANT domain of the EZH2 (enhancer of zeste 2) component. For active genes, the binding of PRC2 to the GQs formed by a nascent RNA likely prevents the engagement of the GQs formed by single-stranded promoter DNA [136]. However, in other situations, the binding of a small RNA to the coding strand would promote GQ formation by promoter DNA without the transcription of a GQ RNA competitor. In this situation, proteins, such as PRC2, that are localized to the site by the small RNA would enhance the formation of a repressive complex at the promoter. In these situations, the small RNA could be produced from a locus elsewhere in the genome [134]. Indeed, small RNAs direct the hiwi (human ortholog of piwi)-mediated repression of human endogenous retroelements in early development and are produced from over 6000 clusters [137,138,139]. By localizing a different set of proteins to the site, small RNAs acting *in trans* could also promote transcriptional activation (Figure 4A). Such a role has been proposed for other piwi-related agonaute family-member complexes [140,141].

### 4.16. R Loop Resolution

A number of mechanisms exist to regulate dGQ formation by R-loops (Figure 3). For example, helicases such as SETX and RTEL1 can facilitate the flip of GQs back to B-DNA through the resolution of RNA:DNA hybrids [142,143]. Nucleases that digest the RNA strand of hybrids, such as RNaseH1, play an important role in their removal [144]. Other proteins such as ATRX prevent R-loop formation at telomeres by sequestering RNA. The deletion of ATRX leads to the increased formation of GQs at telomeres [145].

### 4.17. Chromatin Loops and Transcript Elongation

*In cellulo* studies reveal that delays in RNAP2 transcript elongation occur at the CTCF-binding sites involved in chromatin loop formation. CCTF binds to the large subunit of RNAP2 and the interaction is also associated with cohesin recruitment [146,147,148]. CTCG only binds unmethylated DNA, Consequently, CTCF binding increases following the deletion of the DNA methylase DNMT1.

These findings are consistent with a model where stalling of the polymerase by CTCF results in an R-loop that promotes GQ formation at the site. The GQ structure produced then inhibits DNMT1, preventing DNA methylation of the locus by trapping the enzyme. The trap works because the binding affinity of DNMT1 is higher for GQs than for either duplex, hemi-methylated or single-stranded DNA [149]. The resolution of the GQs by helicases then allows the redocking of the CTCF to the original DNA site, leading to the reinstatement of the chromatin loop formed with the promoter (Figure 4). The CTCF binding sites necessary for the loop transactions lie in reverse orientation to each other. They are then fully aligned at the base of the loop and held in that state until the next round of elongation [85]. After the loop is reestablished, the flipon cycle then resets the DNA locus to be ready for the next transcriptional burst.

### 4.18. Chromatin Loops and Splicing

How GQ formation by DNA affects splicing is therefore of considerable interest. Pausing of RNAP2 is associated with alternative splicing (reviewed in [150]). The sites at which RNAP2 pauses have been investigated at nucleotide resolution. Careful in vivo measurements show a dependence of pause sites on the structure of the RNA:DNA hybrid produced, but not on the canonical DNA motifs that form GQs [151]. The lack of direct involvement of dGQs may reflect the action of the FACT (Facilitated Chromatin Transcription) complex in maintaining the existing epigenetic state by removing nucleosomes in front of the RNAP2 and replacing them behind the enzyme. This mechanism prevents the net accumulation of local DNA supercoiling that might otherwise change flipon conformation [152]. 

However, CTCF-mediated looping is associated with alternative splicing and may allow dGQs to play an indirect role in splicing by maintaining CTCF sites methylation-free. The role for the CTCF is well substantiated. There is evidence that the DNA loops formed between the promoter and the spliceosome mediate the transfer of various splicing factors that initially accumulate in promoter regions [153,154]. There is also ancillary evidence that R-loop formation at promoter sites promotes splicing [155], consistent with the role of GQs in forming promoter/spliceosome condensates.

Alternative splicing is also associated with demethylated DNA, consistent with the role of CTCF-anchored loops in splicing. The deletion of DNMT1 enhances the alternative splicing of the CD45 transcript, as does inducing DNA demethylation, by increasing the expression of TET1 (tet methylcytosine dioxygenase 1) and TET2 enzymes [156,157]. Interestingly, the complement of the degenerate RPOL2 pause motif given by Gajos et al. [151] has a weak match to the CTCF motif (the orientation is inverted relative to those enriched at the TSS). In this case, the inhibition of DNA methylation by GQs may provide a partial explanation for how this conformation can indirectly influence the selection of splice sites [41].

The CTCF-dependent mechanism of connecting promoters with the RNA-processing condensates involved in splicing is quite flexible. For example, the multiple alternative splices of the protocadherin *Pcdh* gene family connect the production of each isoform with a different active promoter [158,159]. A similar dependence on promoter selection is reported for other RNA-processing steps in which the polyadenylation of transcripts occurs at different sites [160,161] (Figure 4). In both outcomes, GQs potentially prevent the loss of CTCF-binding sites by inhibiting the DNA methylation of the locus. The GQ also localizes proteins with roles in their splicing and polyadenylation. The many proposed GQ-binding proteins involved are listed in [63], in the G4IPBD database and in the QUADRatlas database, with a validated subset given in [52].

## 5. RNA and G-Quadruplexes

### 5.1. RNA Modifications and Splicing

rGQs can also form in the RNA transcripts produced, including those with only two tetrads [34] and those folded with non-contiguous G nucleotides [49]. These structures have the potential to alter the RPOL2 elongation rate and the RNA processing performed [41,42]. For example, the splicing factors U2AF65 and SRSF1 bind to GQ RNA with nanomolar affinity, each showing specificity for different GQ substrates [162]. The small molecule cephaeline and the related compound emetine are both reported to impair the formation of GQs by RNA. Both compounds globally disrupt alternative RNA splicing [163]. 

GQ formation may also alter the co-transcriptional N6-methyladenosine (m6A) modification of RNA. It has been proposed that this epigenetic mark can affect splice site selection, but that issue is unresolved [164,165,166]. The involvement of rGQs in m6A modification is also controversial. Interestingly, the methyltransferase METTL3/METTL14 heterodimer that writes m6A within the consensus DRACH motif (D = A, G, or U; R = A or G; H = A, C, or U) binds to rG4 structures preferentially through its RGG domain [167,168]. Also, the RBM15 protein, which also binds rG4, localizes METl3 to certain transcripts and to a subset of H3K36me3 marks [52,166,169]. The mapping of GQs and m6As to splice junctions is dependent on the methods used. Over 81% of GQs that are mapped in HeLa cells are formed from only two tetrads that can stably fold into rGQs [164]. The mapping frequency also depends on the m6A detection protocol employed and the cell line studied, varying from 14% in HeLa cells to 40% in HEK cells [164]. More recent methods are even more sensitive than those used in the earlier analysis, but reproducibility across studies remains a problem [170]. Current mappings do not reveal any enrichment of the DRACH motif in GQ loops, suggesting that rGQs might localize METl3 to modify sequences in their neighborhood [164]. Alternatively, m6A modification may inhibit rGQ formation, as seen for GGA repeats [171]. Interestingly, m6A bases are read by heterogeneous ribonucleic acid proteins (hnRNPs), which are involved in alternative splicing, such as hnRNP C and hnRNP A2B1 [172].

The role of m6A in splicing was also investigated in genetically modified animals. The expression of a hypomorphic METTL3 allele in mouse embryonic stem cells did not appear to change splicing patterns, although there was a slower turnover of many of the wildtype m6A-modified RNAs [165]. Further, in wildtype cells, the distribution of m6A in processed nuclear mRNAs was similar to that found in cytoplasmic mRNAs. Around 70% of the observed m6A sites were in terminal exons, with ~70% in the 3′UTR. Among chromatin-associated RNAs that were not completely processed, ~93% of the m6As in the partially spliced transcripts were in exons and only ~10% of m6As were within 50 nucleotides of 5′ or 3′ splice sites. Notably, methylation was mostly performed before splicing [173]. 

Rather than working with a genomic knockout, another group examined the immediate effects of the acute depletion of METTL3 protein. This approach was designed to minimize the downstream effects on the expression of other genes resulting from METTL3 loss. Around 6–10% of high-confidence m6A regions was mapped to introns, mainly in protein-coding genes, either around stop-codon regions or at the beginning of the 3′UTR. The loss of METTL3 disrupted the inclusion of alternative introns/exons in the nascent transcriptome, particularly at the 5′ splice sites proximal to m6A peaks, suggesting that the sites were occluded or the isoforms were protected by proteins bound to m6A. Among the genes showing altered splicing were those encoding proteins for m6A modification (*Wtap, Ythdc1, Ythdf1 and Spen*), suggesting a negative feedback regulatory mechanism that would be absent in cells with METTL3 deleted from their germline [166]. Overall, the different results for GQ RNA formation at splice sites and METTL3 deficiency are consistent with a model where rGQ-folding in introns can promote the m6A modification of exons, with the rapid degradation of splicing isoforms with retained introns marked by m6A.

### 5.2. Ribosome Assembly

rGQs appear to play an important role in ribosome structure and maturation, with ribosomal RNAs enriched for G-flipons [174]. Many ribosomal proteins have been identified as rGQ ligands in different screens [65,162]. Further, rGQ-binding and resolving proteins such as nucleolin and nucleophosmin help structure the nucleolar condensates that guide ribosome assembly [59,175,176,177]. 

### 5.3. Translation

rGQ formation by mRNA is the subject of much interest, especially in the untranslated regions that regulate translation. These exons contain alternative translation initiation sites and microRNA (miR)-binding sites that affect the production of different protein isoforms. The complexities involved are described in a number of recent reviews. These articles provide examples of how rGQs in 5′UTRs can switch the use of start codons to produce completely different protein products, while rGQs in the 3′UTR can modulate the translation of mRNAs and interactions with small regulatory RNAs such as miRNA [178,179,180,181]. An analysis of G-flipons in 5′- and 3′UTR provides evidence of positive selection, which can alter the alternative splicing of these exons. Single-nucleotide variants in both 5′- and 3′UTR are associated with quantitative trait loci [182]. Bioinformatic approaches have also been used to identify G-flipon RNA-binding protein, as annotated in the QUADRatlas database.

By modulating mRNA translation RNAs, rGQs contribute in many ways to phenotypic pliability [28]. Here, helicases such as DHX36 and CCHC-type zinc-finger nucleic acid-binding protein (CNBP/ZNF9) play a central role in promoting mRNA translation by resolving rGQs [183,184]. The m6A modifications of RNA that are associated with rGQ formation during transcription (as described above) also impact translation. The removal of these marks from the 5′UTR near the start codon by the m6A erasers AlkB homolog H5 (ALKBH5) and fat mass and obesity (FTO) decreases ribosome translational pausing, increasing protein synthesis [185]. Such m6A modifications also dynamically regulate heat-shock responses by enhancing N7-methylguanosine cap-independent translation [186]. Further, the class I cytoplasmic m6A readers YTHDF1 and YTHDF3 promote the degradation of target transcripts [187], potentially eliminating partially processed transcripts with retained introns. The endogenous repeat elements present in these introns, such as ALU SINE inverted repeats, might otherwise activate dsRNA- and Z-RNA-dependent immune responses [130]. The potential of rGQs to enhance m6A modifications provides additional mechanistic insight into how G-flipons increase phenotype pliability by regulating RNA-dependent epigenetic outcomes.

## 6. Flipons and Development

### 6.1. Pioneering Factors and Flipons

Other mechanisms exist for the induction of alternative flipon conformations. Sequence-specific pioneering transcription factors, such as HNF4 and GATA4, can dock to their motifs on nucleosome-bound DNA. The master-regulators of embryonic development then localize complexes that evict histone octamers from the locus, generating a negatively supercoiled NDR at the site [188,189]. The energy released by the removal of a nucleosome is sufficient to induce a number of different alternative DNA conformations [190]. The relaxation of these structures to B-DNA is sufficient to power the assembly of the different biological machines that actuate alternative cellular responses (Figure 3). 

### 6.2. Bootstrapping Flipon Conformation with Noncoding RNAs

GQs are able to facilitate a number of different processes in the cell that are directed by sequence-specific TFs. Small noncoding RNAs, such as those used in the piwi system to regulate endogenous retroelements [191], provide another means by which GQ formation can be regulated in a sequence-specific manner. In both cases, the alternative flipon conformations engage the same structure-specific cellular machinery. The question arises as to how these two different systems for sequence-specific regulation of gene expression and RNA translation are used to program development, especially during early embryogenesis. To explore the role of small RNAs in this process, the sequence-specific match between experimentally confirmed flipons and miR highly conserved in eutherian mammals was explored. Intriguingly, promoters with miR matches to G- and Z-flipons were highly enriched in developmental genes (FDR > 10^−100^), The findings are consistent with a role for miRNA in programming flipon conformation during early embryogenesis [134].

Notably, GQs are enriched in human embryonic stem cells (hESCs). About 18,000 GQs were mapped to the NDR, as defined by ATAC seq. Following differentiation into neural stem cells and cranial neural crest cells, the number of detectable GQs was reduced by 25–50%, with findings differing by lineage [192]. In hESCs, GQs were mapped to ~50% of bivalent promoters that contain both active H3K4me1 and repressive H3K27me3 marks, and were lowly transcribed. The GQs in hESCs overlapped sites bound by the CTCF (~36%), the cohesin component RAD51 (~50%) and RING1B, which mediates repression by recruiting PRC1 to R-loops (~55%) [193]. Differentiation was associated with the loss of bivalent promoters, reflecting the potential of GQs to localize either activating or repressive protein complexes during lineage specification. Collectively, the results are consistent with a model where small RNAs bootstrap development, much in the same way a computer loads an initial program to specify the inputs and outputs that are necessary for an operating system to run. Here, the programming of flipon conformation by small RNAs would establish epigenetic marks to template tissue differentiation by sequence-specific B-DNA-binding proteins. The bootstrapping by small RNAs that occurs after the erasure of existing parental epigenetic marks early in development could potentially involve miR, transmitted by either maternal or paternal gametes [194,195,196,197]. Further research is needed to address such mechanisms. 

## 7. Summary and Outlook

Flipons are genetically encoded elements that dynamically change their conformation under physiological conditions without requiring strand cleavage or a change in sequence. They vary by the non-B-DNA structures they form. Z-flipons flip rapidly, with an in vitro relaxation time of 100 ms, and have ancient, well-documented roles in self-recognition and immunity through their structure-specific interactions with the Zα domain [130]. G-flipons are much more stable, with higher melting temperatures than their B-DNA structure and the potential to form bistable switches. Yet, like Z-flipons, GQs are formed and resolved dynamically to perform a number of important biological roles (Figure 3). Flipons that form triplexes are also likely to influence gene expression and development [198,199], with examples related to the hemoglobin locus [200] and to triplex stabilization by histone H3 tails [201,202]. Notably, the Drosophila GAGA protein binds triplex-DNA through the same domain that binds B-DNA in a sequence-specific manner [203]. Triplex-forming sequences are also enriched in repeat elements, such as ALU SINEs (short interspersed nuclear elements), which form part of the repetitive genome [106]. Other triplexes are formed by long noncoding RNAs. Their biology then reflects the RNA motifs that the triplex forming sequences deliver to a locus. The sequence- and structure-specific proteins engaged by the tag along motifs then scaffold the formation of various chromatin-modifying complexes [204].

Based on a dynamic form of encoding, flipon biology can be best visualized as a cycle that exchanges energy for information. The flip to an alternative conformation is regulated both genetically and by environmental events, by base modifications that enhance or suppress the transition. The outcomes depend upon proteins and noncoding RNAs that modulate the formation or resolution of their alternative conformation. These modulators are themselves subject to modifications that help tune the cycle. Other factors also affect the flipon equilibrium by binding in a sequence-specific manner to right-handed B-DNA conformations or to single-stranded RNA to oppose the flip to the alternative conformation. 

While it has been usual to consider the effects of evolution on the individual protein components involved in cellular processes, the optimization of so many different parameters represents a combinatorically challenging calculation full of cascading complexity, similar in logic to the epicycles once used to predict planetary orbits in a bygone era. Instead, flipons offer a simpler alternative to optimize context-specific responses that allow rapid adjustments of the cellular state in response to environmental perturbations. By programming and refreshing the epigenetic state of a cell, flipons facilitate the formation and maintenance of cellular memory [2]. Here, the various ways in which G-flipons impact a wide variety of biological processes are described, with a focus on the recent experimental validations of GQs and descriptions of what is currently unknown.

## Figures and Tables

**Figure 1 ijms-25-10299-f001:**
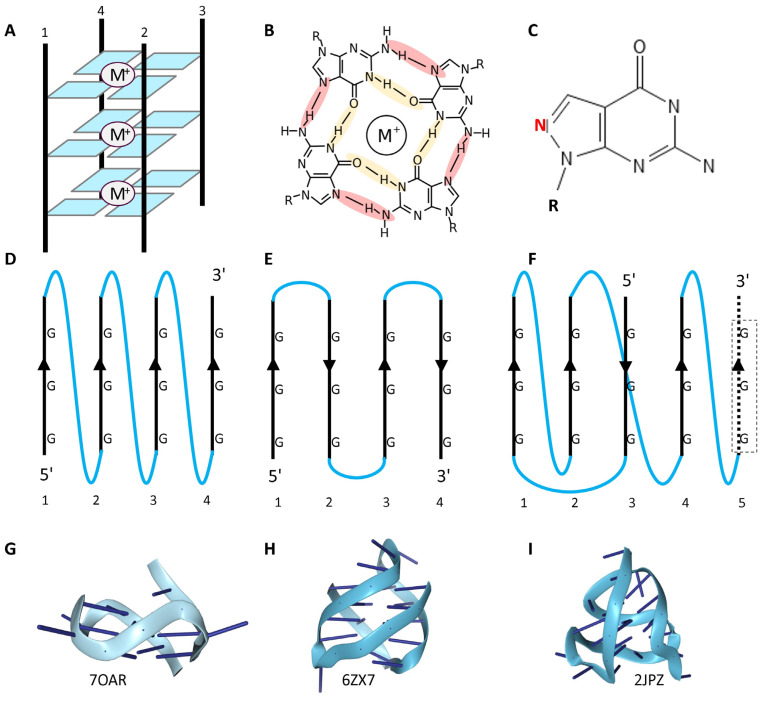
**GQ fold in many different ways**. (**A**) The core four-stranded structure formed by stacking the guanine tetrads shown in (**B**), with Hoogsteen hydrogen bonds highlighted in yellow and crimson. The four strands may form from G-repeats on the same molecule, arise from different molecules or arise from either RNA or DNA. (**C**) The base 8-aza-7-deazaguanosine retains the same molecular composition as guanosine, but with the red ring nitrogen in a different position, preventing the formation of the Hoogsteen hydrogen bonds shown with crimson shading. Control oligonucleotides incorporating this nucleotide will not form GQs. In intramolecular GQs, the stands may be parallel (**D**,**G**), anti-parallel (**E**,**H**) or hybrid (**F**,**I**). The topology of the connecting loops is shown in blue and can be propeller (**D**), lateral (**E**) or diagonal (**F**). M^+^ indicates a metal ion located at the core of the tetrad. K^+^ promotes GQ formation while Li^+^ does not. The dotted strand in (**F**) labeled 5 indicates that many sequences capable of forming GQs contain a “spare” tire that can maintain the fold when one of the other repeats is damaged [37]. The cartons in (**G**–**I**) show the phosphate backbone as a ribbon and the bases as sticks. PDB codes are given below the structures.

**Figure 2 ijms-25-10299-f002:**
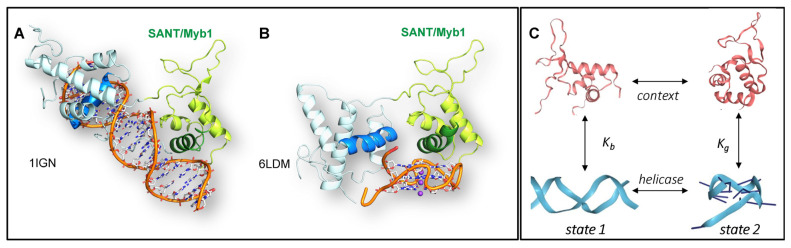
**Some proteins bind both B-DNA and GQs**. The yeast Rap1 protein binds both to B-DNA in a sequence-specific manner (**A**) [55] and to GQs in a structure-specific manner (**B**) [56] through different faces of the same helix in the SANT/Myb1 domain (Images by Daniela Rhodes) (**C**). The flipon cycle creates a two-state switch with similar affinities of Rap1 for B-DNA and for GQs (i.e., K_b_ ≈ K_g_ ≈ ~20–30 nM). Flipons thereby enable binary coding within the genome.

**Figure 3 ijms-25-10299-f003:**
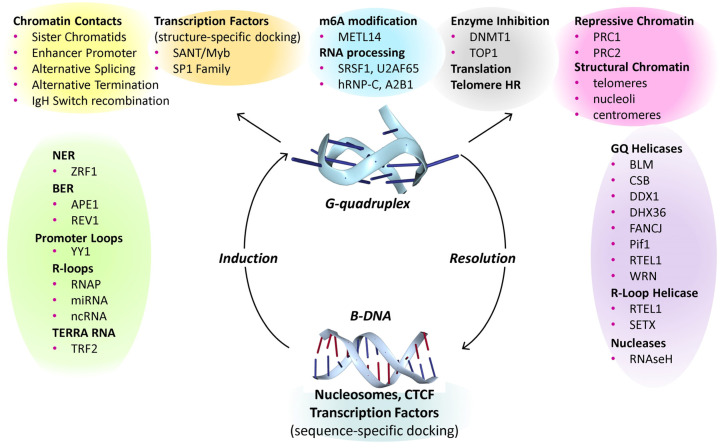
**The G-flipon cycle**. Many factors induce and resolve GQs to modulate specific outcomes. Other proteins prevent their formation, such as nucleosomes and repressor proteins. Transcription factors can dock to G-flipons in either a sequence-specific manner to their right-handed conformation or to the GQ structure. Once formed, the resolution of GQs can be coupled to the different outcomes shown, with both activation and inhibition of gene expression. The inhibition of enzymes like DNMT1 and TOP1 favors the maintenance of an unmethylated, nucleosome-depleted state that is necessary to rapidly reprogram cellular responses to environmental inputs. (BER: base excision repair; NER: nucleotide excision repair). References for each gene are given in the text.

**Figure 5 ijms-25-10299-f005:**
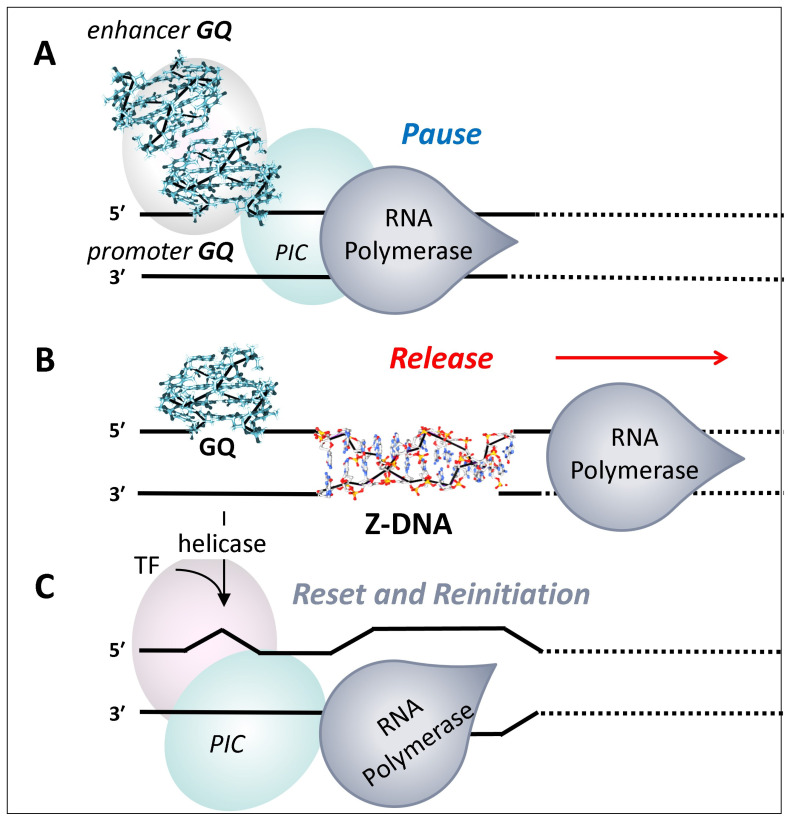
**Flipon-cycle promoters**. The reset and reinitiation of transcription complexes are actuated by G- and Z-flipons. (**A**) In this model, a condensate is anchored by GQs formed by enhancer and promoter sequences. The condensate stabilizes an enhancer–promoter loop and holds the RNA polymerase (RPOL2) in a poised state. (**B**) Breakdown of the condensate triggers transcript elongation. The negative supercoiling 5′ to RNAP2 induces Z-DNA formation that actuates the removal of the pre-initiation complex (PIC). (**C**) The resolution of the promoter GQ by helicases then enables the rebinding of TFs (transcription factors) and reassembly of the PIC. The separation of strands indicates the partial unwinding that is necessary to form the transcription bubble and a GQ.

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
