# Peer review of "A Compendium of G-Flipon Biological Functions That Have Experimental Validation"

_ijms, 2024, doi:10.3390/ijms251910299_

Round 1

Reviewer 1 Report

Comments and Suggestions for Authors

In his review entitled “A Compendium of G-flipon biological functions that have experimental validation”, the author, Alan Herbert has present a description of almost all of the information concerning biological roles of G-quadruplexes.

G-quadruplexes (GQ) are quite interesting DNA structures that must be thoroughly studied because of their proven and suspected roles in important biological processes. G-quadruplexes are involved in the norm as well as in the pathology of these processes. Therefore, a review on the subject could be very useful for the specialists in the field and the general public.

General

The manuscript reviews almost all of the data and is quite informative and detailed, giving deep insight into the possible mechanism of GQ influence on genome activity, replication and repair.

Pros

The author is one of the not-so-many specialists in the field.  Therefore, he brought together all of the available information on GQs. This is a real Compendium. Moreover, he critically discussed possible mechanisms of GQs’ action. It is not easy to join and represent sometimes contradictory data on GQs. However, the author succeeds in the fulfilment of this task.

Cons

There are not many things which are not properly done in this review.

I have found several typos and some improperly constructed sentences like:

-  The canonical DNA motif (G3N1-7)3G3, where N is any nucleotide and G is guanine, a feature that is under active selection in avian and mammalian genomes. (lines 10-11).

- The mappings show that In the human….. (line 110).

-          etc.

My concise is the manuscript to be re-read by the author and all of these minor mistakes to be fixed.

Decision

This is a very detailed and concise description of the G-quadruplexes and their roles. The review will undoubtedly be met with interest from the specialists in the field and the scientific community as a whole.

Therefore, I recommend this review be considered for publication in IJMS in the present form.  

Author Response

Thanks for the great feedback.

I have rewritten much of the review and added two new figures and updated the 3 original figures and added 34 additional references. The aim was to add further clarity to the review.

In the process, many typo's have been fixed.

Reviewer 2 Report

Comments and Suggestions for Authors

In this study, the author provides a comprehensive review on the biological function of G-quadruplexes (GQ)s.

Overall, the manuscript is interesting, and the topic is properly described and discussed. 

In my opinion, the first part of the paper (including “Introduction” and “Biophysical and Computational Studies of the G-quadruplex”) could be improved by a more fluid and simple description of G-flipons, possibly accompanied by further illustrations to make the topic more understandable even for non-experts who are interested in the second part of the manuscript, which details the biological importance of GQs.

On the other side, the second part should be better reorganized through the use of sub-headings that group together several related processes (such as promoter maintenance, resets, enhancer etc)

Other comments:

Figure 3 could be recalled in further points of the text.

Some sentences are interrupted (see for example lines 207-209).

Author Response

Thanks-you for  your thoughtful review.

  1. In my opinion, the first part of the paper (including “Introduction” and “Biophysical and Computational Studies of the G-quadruplex”) could be improved by a more fluid and simple description of G-flipons, possibly accompanied by further illustrations to make the topic more understandable even for non-experts.

 I have extensively rewritten the paper to address the points you raise and add new figures  and updated the 3 original figures.

The introduction of flipons is now extended in the introduction, in the section of biological relevance and in the discussion. I have added to figure 3 to show a flipon cycle based on Rap1 and how this enable a two-state switch.

I have added figure 4 to show transactions between GQ forms at different genomic locations that modulate gene expression and mRNA processing, and provided a retroviral example of how the condensates can be used to regulate gene expression. In figure 5, I have tried to illustrate the flipon cycle associated with promoter reset that involves both G- and Z-flipons.

  1. On the other side, the second part should be better reorganized through the use of sub-headings that group together several related processes (such as promoter maintenance, resets, enhancer etc.)

Thanks for the suggesting. The section has been completely reorganized with sub-headings to better index the subjects discussed.

  1. Figure 3 could be recalled in further points of the text.

The figures are now all better referenced in the text.